**Data Availability Statement:** All relevant data are within the manuscript and its Supporting Information files.

# The clinical course of hospitalized moderately ill COVID-19 patients is mirrored by routine hematologic tests and influenced by renal transplantation

**Paula M. Peçanha-Pietrobom**[1], **Giuseppe Gianini Figueirêdo Leite**[1], **James Hunter**[1], **Paulo R. Abrão Ferreira**[1], **Marcelo N. Burattini**[1], **Nancy Bellei**[1], **Jaquelina Sonoe Ota-Arakaki**[2], **Reinaldo Salomao**[1]*

1 Division of Infectious Diseases, Escola Paulista de Medicina, Universidade Federal de Sao Paulo, Sao Paulo, Brazil, 2 Division of Respiratory Diseases, Escola Paulista de Medicina, Universidade Federal de Sao Paulo, Sao Paulo, Brazil

* rsalomao@unifesp.br

## Abstract

Several studies of patients with COVID-19 have evaluated biological markers for predicting outcomes, most of them retrospectively and with a wide scope of clinical severity. We followed a prospective cohort of patients admitted in hospital wards with moderate COVID-19 disease, including those with a history of kidney transplantation, and examined the ability of changes in routine hematologic laboratory parameters to predict and mirror the patients' clinical course regarding the severity of their condition (classified as critical vs. non-critical) and in-hospital mortality or hospital discharge. Among the 68 patients, 20 (29%) were kidney transplanted patients (KT), and they had much higher mortality than non-kidney transplanted patients in this cohort (40% X 8.3%). Lymphocytes, neutrophils and neutrophils/lymphocytes ratio (NLR) at admission and platelets as well as the red blood cells parameters hemoglobin, hematocrit, and RDW by the time of hospital discharge or death clearly differentiated patients progressing to critical disease and those with clinical recovery. Patients with deteriorating clinical courses presented elevated and similar NLRs during the first week of hospitalization. However, they were dramatically different at hospital discharge, with a decrease in the survivors (NLR around 5.5) and sustained elevation in non-survivors (NLR around 21). Platelets also could distinguish survivors from non-survivors among the critical patients. In conclusion, routine hematologic tests are useful to monitor the clinical course of COVID-19 patients admitted with moderate disease. Unexpectedly, changes in hematologic tests, including lymphopenia, were not predictive of complicated outcomes among KT recipients.

**Funding:** This work was supported by Fundacao de Amparo a Pesquisa do Estado de Sao Paulo (FAPESP) – Grant number 2017/21052-0 and 2020/05110-2 to RS.

**Competing interests:** The authors declare that they have no competing interests.

## Introduction

In December 2019, the city of Wuhan became the center of an outbreak of pneumonia of unknown origin, rapidly identified as caused by a new coronavirus, the SARS-CoV-2 (from "severe acute respiratory syndrome coronavirus 2"). The disease was characterized as COVID-19 (an acronym for "coronavirus disease") by the World Health Organization (WHO) on February 11th, 2020, and in just one month, on March 11, was declared as a global pandemic [1].

The first case in Brazil was reported on February 26th, 2020. The travel history and genetic analysis of the virus confirmed that the infection was imported from Northern Italy [2]. Worldwide cases updated in September 2021, exceed 220 million cases leading to 4.6 million deaths of which 585 thousand occurred in Brazil [3].

After more than one year from the description of the first case, the world still faces great challenges in the management of COVID-19 patients mainly among fragile populations. There are emerging new data on the clinical presentation of COVID-19 in specific populations, such as transplant patients. Organ transplantation has been associated with increased risk for COVID-19-related deaths, but the role of age, comorbidities and immunosuppression are still not clear [4,5]. The university affiliated Hospital São Paulo (site of this study) treated 1658 cases from March 2020 to April 2021 of whom 498 passed away (a mortality rate of 30,03%). In the first months of the pandemic (between March and July 2020), our affiliated kidney transplant center had observed 491 patients with Covid-19 among the 11,8975 kidney transplants (KT) recipients who were in follow-up in that period (4.1%). Thirty-one percent of these patients were treated at home. Among the hospitalized, 61% needed intensive care and 41% died [6].

The decision to monitor a patient in the inpatient or outpatient setting depends on the clinical presentation, requirement for supportive care, potential risk factors for severe disease, and the ability of the patient to self-isolate at home [7]. Guidelines in Brazil recommend hospital admission for patients with a respiratory rate $\geq 24$ / min and peripheral arterial oxygen saturation (SpO2) $\leq 94\%$ [8]. Those with critical disease (respiratory failure, shock, or multiple organ system dysfunction), who represented 5% of patients in a large China cohort, should be referred to ICU [9]. Patients admitted in wards, usually with moderate disease, might have distinct clinical courses, leading to clinical and laboratory improvement and hospital discharge (non-complicated), or leading to clinical deterioration, critical disease (complicated) and eventually death. Considering the wide spectrum of clinical presentation, the WHO stratified the COVID-19 disease according to the severity in COVID-19 disease as mild, moderate, severe and critical disease [10].

Several studies of patients with COVID-19 evaluated biological markers for predicting outcomes [11,12]. Besides the presence of underlying diseases and age, higher Sequential Organ Failure Assessment (SOFA) score, elevated d-dimer concentrations, lymphopenia, and elevated cytokines levels relate to severe cases and unfavorable outcomes [13–15].

It is worth highlighting that most studies evaluating biomarkers are retrospective, having enrolled a wide range of hospitalized patients. Several of them are based on admission blood samples [16]. Here, we focused on patients with moderate disease, including patients with a history of kidney transplantation, who were regularly admitted in hospital wards. We followed them prospectively, studying their clinical course during hospitalization regarding the severity of their condition (classified as critical vs. non-critical), need for ICU support and in-hospital deaths, examining the potential role of changes in routine hematologic laboratory parameters to predict and monitor these trajectories.

## Methods

This is a prospective cohort study carried out at Hospital São Paulo, a tertiary university hospital in the city of São Paulo, with approximately 12.3 million inhabitants. The study protocol was approved by the research ethics committee and all volunteers gave a written informed consent before enrollment in the cohort (Process number 4.453.137). Inclusion criteria were age over 18 years and need for hospitalization due to clinical symptoms. The diagnoses of COVID-19 were confirmed by the detection of SARS-CoV-2 RNA by polymerase chain reaction (PCR) in a nasopharynx swab.

Data were prospectively collected in a RedCap database (Research Electronic Data Capture, Vanderbilt University, US, hosted by Escola Paulista de Medicina). Data collected at admission included age, sex, pregnancy, time since the initiation of symptoms, the symptoms experienced by the patients, contact with SARS-CoV-2 infected individuals, comorbidities, and patient's body mass index (BMI). Heart rate, systolic and diastolic systemic blood pressure, respiratory rate, body temperature, Glasgow coma scale, and SpO2 were collected daily in the wards. For those transferred to the intensive care unit, we also collected data on the need of respiratory support and vasoactive drugs. Associated infections were screened throughout the follow-up.

Routine hematologic tests were performed at inclusion, day three, day seven of hospitalization and at discharge, including: hemoglobin, hematocrit, red cell distribution width (RDW), leukocytes, neutrophils, lymphocytes, monocytes and platelets. Other laboratory tests included blood gas analysis, creatinine, C-reactive protein, d-dimer, ferritin, and lactate. After informed consent, blood samples were also collected as part of ongoing studies to evaluate cellular dysfunction, biomarkers, and proteome changes (FAPESP Grant number 2020/05110-2).

Chest computed tomography was performed at admission and accordingly to the assistant discretion.

The SOFA score was calculated at inclusion, day three and day seven [17]. Comorbidities were assessed using the Charlson comorbidity index [18]. Drug treatments, both before and during hospitalization, are described.

The outcomes were the clinical course during hospitalization (classified as critical vs. non-critical), need for ICU support and in-hospital mortality.

### Statistical analysis

Statistical analysis was done with SPSS Statistics Software (version 21; IBM Corp, New York, USA). Student's t-test was used to compare the normally distributed continuous variables while Mann–Whitney U-test was used to compare non-normally distributed continuous variables, and categorical data were compared with the chi-squared test. P-values < 0.05 were considered statistically significant. Principal Components Analysis (PCA) was employed in order to identify the set of variables that most correlates with the desired outcomes. PCA intends to reduce the number of variables influencing the outcomes by identifying others that are most related to them. It analyzes variance-covariance matrices aiming to detect those variables (even unobservable ones) that most correlate with the desired outcomes have the greatest overall correlation (thus the name principal components). As here, the process of discovering new variable components assists in identifying those variables that are most important to the determination of the result.

Graphics were prepared using the ggplot2 package (of the R Language and Environment for Statistical Computing (version 4.0.5, R Foundation for Statistical Computing, Vienna Austria).

## Results

### Demographic and clinical characteristics at hospital admission

This cohort consists of 68 patients, 20 of them KT recipients, who presented with moderate disease and were admitted to the hospital wards between May 10th and September 26th, 2020.

Kidney transplants were performed between 1996 and 2020. Seven patients were transplanted less than one year before hospitalization, one patient between one and five years, and twelve more than 5 years before the COVID event. Maintenance immunosuppressive therapy mycophenolate (N = 17), tacrolimus (N = 16), azathioprine (N = 3), cyclosporine (N = 2). All patients were under prednisone treatment.

The mean age of the COVID-19 patients was 57.3 years (SD ± 12.6) and 61.8% were male. The most common symptom at admission was cough, observed in 73.5% of the patients. Fever was present in 70.6%, shortness of breath in 67.6% and diarrhea in 30.9% of patients. In the full cohort, 94% had at least one coexisting illness and 29% were KT recipients. The mean Charlson comorbidity index was 3.32 (SD 2.14), with hypertension being the most common comorbidity (63.2%), followed by diabetes (39.7%). At admission, mean respiratory rate was 24 / min (SD±5.6) and mean SpO2 was 93%. (S1 Fig)

The average SOFA score was 1.95 (SD±1.75). Chest CT (computed tomography) scan performed in all patients at admission revealed 25%-50% lung involvement in most patients (71%), with 14.5% presenting less than 25% and 14.5% showing more than 50% involvement.

### Clinical course and routine hematologic tests changes

The median time from symptoms' onset to hospital admission was 7.00 days (Interquartile Range (IQR): 5.00–9.25). The clinical condition of 24 patients impaired during the study period, with twenty-two patients being transferred to the ICU. The median time from hospital admission to ICU transfer was 3.50 days (IQR: 2.00–4.75), with a median stay in ICU of 14.0 days (IQR: 10.0–31.0). Fifteen patients (22.1%) received mechanical ventilation support after a median time from hospital admission of 6.0 days (IQR: 4.0–9.00) and with a median time of ventilatory support of 14.0 days (IQR: 10.5–29.0). Twelve patients (17.6%) died during the study period. Demographic and clinical characteristics are presented in Table 1 with patients divided accordingly to the clinical course.

Routine hematologic laboratory results at admission revealed mean lymphocytes cell counts lower than 1000 cells/μl in patients who deteriorated to critical disease (p = 0.043 compared to the non-critical group). Also, a higher neutrophil cell counts and a striking difference in the ratio of neutrophils to lymphocytes (NLR) occurred in patients whose conditions became critical, whether they survived or not, as compared to those with clinical recovery (Fig 1, S1 Table).

PCA model measuring the difference in values for each of the laboratory tests between day 1 and day 3 created dimensions with a clear distinction among the laboratory tests. In this model, lymphocytes and monocytes are both about equal strength and direction and contrast with neutrophils, which are of approximately the same strength but in a different direction (of the balance between the first two dimensions) than the first two variables. Creatinine and RDW are also relatively strong measures (S2 Fig).

As patients progressed to a more severe form of the disease, routine hematologic results were more distinct among the groups. During the first week of disease, lymphocytes cell count increased from admission (D0) to D3 and D7 in patients with uncomplicated disease and remained low in patients with critical disease (Fig 1A, S1 Table). In contrast, neutrophil cell counts increased in patients with a complicated clinical course and remained stable in those whose condition did not become severe (Fig 1B, S1 Table). The NLR, therefore, was

**Table 1. Epidemiologic and clinical characteristics of the cohort accordingly to the clinical course and outcomes of the disease.**

| | Critically ill (N = 24) | | Non-critical (N = 44) | P-Value * | P-Value ** |
|---|---|---|---|---|---|
| | Hospital discharge (N = 12) | In-hospital death (N = 12) | | | |
| **Demography** | | | | | |
| Male sex, n (ratio) | 6 (0.50) | 7 (0.58) | 29 (0.65) | 0.682 | 0.341 |
| Age, Mean (SD) | 64.13 (9.0) | 59.00 (9.4) | 55.16 (13.6) | 0.188 | 0.058 |
| **Admission data** | | | | | |
| Day of symptoms | 5.8 (1.8) | 6.9 (3.5) | 8.2 (4.4) | 0.365 | 0.109 |
| Fever, n (ratio) | 7 (0.58) | 8 (0.66) | 33 (0.75) | 0.673 | 0.281 |
| Cough | 8 (0.66) | 8 (0.66) | 34 (0.77) | 1 | 0.343 |
| Shortness of breath | 10 (0.83) | 9 (0.75) | 27 (0.61) | 0.615 | 0.134 |
| Diarrhea | 6 (0.50) | 4 (0.33) | 11 (0.25) | 0.408 | 0.155 |
| Temperature, Mean (SD) | 36.3 (1.2) | 36.8 (1.2) | 36.1 (0.9) | 0.68 | 0.090 |
| Cardiac Rate | 83.8 (15.2) | 85.1 (13.0) | 88.4 (15.5) | 0.82 | 0.305 |
| Respiratory rate | 22.0 (2.8) | 24.9 (5.7) | 24.5 (6.0) | 0.171 | 0.733 |
| SpO2 | 93.3 (2.9) | 90.8 (3.5) | 93.1 (3.4) | **0.042** | 0.180 |
| Body mass index | 26.8 (5.2) | 25.6 (2.2) | 28.9 (6.8) | 0.502 | 0.118 |
| SOFA score | 1.8 (1.7) | 3.3 (2.4) | 1.6 (1.3) | 0.095 | 0.082 |
| **Comorbidities** | | | | | |
| Cardiac disease, n (ratio) | 1 (0.08) | 2 (0.16) | 10 (0.22) | 0.537 | 0.305 |
| Chronic pulmonary disease | 2 (0.16) | 1 (0.08) | 6 (0.13) | 0.537 | 0.895 |
| Diabetes | 6 (0.50) | 5 (0.41) | 16 (0.36) | 0.682 | 0.446 |
| Chronic kidney disease | 4 (0.33) | 9 (0.75) | 11 (0.25) | **0.041** | **0.016** |
| Hypertension | 8 (0.66) | 9 (0.75) | 26 (0.59) | 0.653 | 0.337 |
| Obesity | 1 (0.08) | 0 (0.0) | 7 (0.15) | 0.307 | 0.151 |
| Stroke | 1 (0.08) | 0 (0.0) | 3 (0.06) | 0.307 | 0.657 |
| Charlson Comorbidity index, Mean (SD) | 4.2 (2.1) | 4.4 (1.7) | 2.0 (2.7) | 0.948 | **0.003** |
| Hospital days | 30.0 (21.5) | 31.6 (22.5) | 8.2 (5.8) | 0.795 | **<0.0001** |

dramatically higher in D3 and D7 in severely ill patients (survivors and non-survivors) than in those with an uncomplicated clinical course (Fig 1C, S1 Table).

Samples obtained at hospital discharge showed differences between patients with critical disease and those with clinical recovery in most evaluated hematologic parameters, including the red blood cell parameters hemoglobin, hematocrit, and RDW (Fig 1A–1E, S1 Table). Except for lymphocytes, they also differed between survivors and non-survivors among those critically ill (S1 Table).

Among other routine laboratory blood tests, creatinine levels differed among the patients progressing to a critical disease and those with clinical recovery at D3, D7 and at hospital discharge. The hospital discharge samples showed that patients with critical disease who survived had lower levels of creatinine than those who died (S1 Table). C-reactive protein (CRP) levels tended to be higher in patients with more severe disease but reached significance only in hospital discharge samples (S1 Table). Ferritin and D-dimer analysis were prejudiced by the high number of missing values during follow-up. Lactate at admission did not differ statistically among the groups and the follow-up was hampered by missing values.

## KT recipients' effects on cohort outcomes and laboratory findings

A direct comparison between KT (N = 20) and non-KT (N = 48) patients shows that the clinical symptoms were similar for both groups at admission, with respiratory rate and SpO2 in the

# Evolution of Hemogram During Treatment

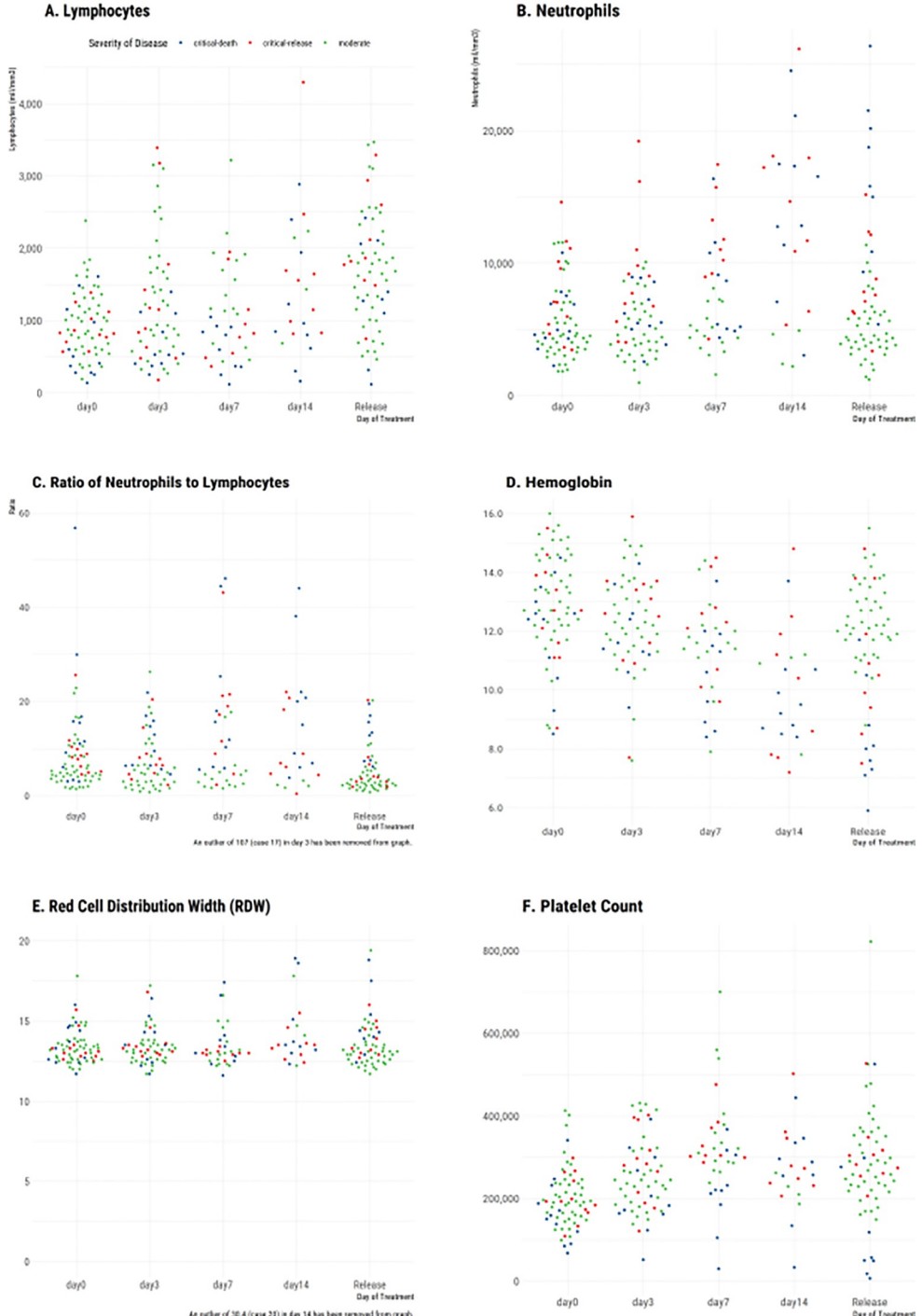

**Fig 1. Hematologic laboratory tests segregated accordingly to the clinical course (moderate and critical) and outcomes (survivors and non-survivors).** Lymphocytes (A), neutrophils (B), neutrophils to lymphocytes ratio (C), hemoglobin (D), RDW (E), and platelets (F).

same range (Table 2). KT patients tended to be younger and had a higher SOFA score at admission than non-KT recipients. Chronic obstructive pulmonary disease (COPD) was more common in non-KT patients, the same trend as cardiac disease, while hypertension was more common in KT recipients. All KT patients were considered as having chronic kidney disease (CKD), as defined in KDIGO clinical practice guideline for the evaluation and management of chronic kidney disease [19]. The mortality rate was much higher in KT recipients than in non-KT (40%—or 8 out of 20 versus 8.3%—or 4 out of 48, p = 0.002). Peripheral blood cell tests showed lower lymphocytes cell counts in KT at all time points, lower monocyte counts at D3 and higher NLR in follow-up samples (D3, D7 and hospital discharge). Red blood cell parameters show a slightly lower Hb levels at admission and at hospital discharge in KT than in non-KT patients, respectively. RDW was distinct only in hospital discharge measurement (Table 2).

Among the 48 non-KT COVID-19 patients, 14 (29.2%) had a deteriorating clinical course, among whom four (8.3%) died during hospitalization. CKD was more common and the Charlson index was higher among those patients whose condition worsened during hospitalization. The length of stay in the hospital was much longer for those patients with a complicated clinical course (over 30 days) than for those with an uncomplicated clinical course (around 8 days) (S2 Table). Mean lymphocyte cell counts were lower than 1000 cells/μl in patients whose condition deteriorated and higher in uncomplicated patients at admission (S2 Table). At day 7, patients who later died presented lower lymphocytes cell counts than those who survived (Fig 2). Neutrophils and the NLR were higher at admission and at D3 and D7 in patients with a complicated course. Monocytes differed only at hospital discharge with higher cell counts in patients with severe disease than in uncomplicated ones (S2 Table). Red blood cell parameters were not distinct among the groups at any time point, except for RDW at hospital discharge. Creatinine levels at admission were similar between complicated and uncomplicated patients, with higher levels in complicated patients at hospital discharge. CRP levels were higher in the complicated group from the third day until hospital discharge.

KT recipients presented clinical deterioration in 50% of the cases, and 8 out of 10 patients with a complicated course died (S3 Table). Length of stay was 30.3 (SD = 16.7) days for those with a complicated clinical course and 9.7 days (SD = 6.2) for those with clinical recovery. Comorbidities and clinical presentation were similar at admission, with a trend of lower SpO2 in severe patients. Lymphocyte cell counts were lower than 1,000 cells/μl in both groups at admission and on days 3 and 7, without difference between complicated and non-complicated groups (S3 Table). In contrast to non-KT recipient patients, there was no difference of lymphocytes cell counts on D7 between survivors and non-survivors (Fig 2). At hospital discharge, neutrophils and NLR were higher in complicated cases than in those with an uncomplicated clinical evolution. The same was true for hemoglobin and hematocrit. At day 7 and hospital discharge, creatinine levels were higher in patients with critical illness than in those uncomplicated patients (S3 Table).

## Discussion

In this prospective cohort of moderately ill COVID-19 patients, routine hematologic tests were a useful bedside tool to monitor patients with distinct clinical courses.

The wide scope of disease severity and outcomes of COVID-19 appeared early in the pandemic and remains a challenge for clinical management and research after a year and a half. Thus, COVID-19 patients may present from asymptomatic to critical disease, with mortality rates in patients admitted to the ICU ranging from 39% to 72% [7].

We focused on the clinical course of patients admitted to the hospital wards with moderate disease, aiming to identify changes early in the disease course that could predict or indicate a

**Table 2. Epidemiologic and clinical characteristics, and routine laboratory tests of KT recipients and non-KT recipients COVID-19 patients.**

| | SOT (N = 20) | No SOT (N = 48) | P-Value [**] |
|---|---|---|---|
| **Demography** | | | |
| Male sex, n (ratio) | 12 (0.60) | 30 (0.62) | 0.847 |
| Age, Mean (SD) | 54.6 (10.8) | 58.6 (13.2) | 0.111 |
| **Admission data** | | | |
| Day of symptoms | 8. 0 (4.8) | 7.4 (3.6) | 0.916 |
| Fever, n (ratio) | 16 (0.80) | 32 (0.66) | 0.272 |
| Cough | 14 (0.70) | 36 (0.75) | 0.670 |
| Shortness of breath | 15 (0.75) | 31 (0.64) | 0.635 |
| Diarrhea | 7 (0.35) | 14 (0.29) | 0.635 |
| Temperature, Mean (SD) | 36.6 (0.8) | 36.8 (1.1) | 0.580 |
| Cardiac Rate | 85.0 (12.8) | 87.8 (15.7) | 0.489 |
| Respiratory rate | 23.6 (3.5) | 24.3 (6.2) | 0.710 |
| SpO2 | 93.0 (2.9) | 92.6 (3.6) | 0.973 |
| Body mass index | 25.4 (4.0) | 29.0 (6.5) | 0.084 |
| SOFA score | 2.7 (1.7) | 1.6 (1.6) | **0.010** |
| **Comorbidities** | | | |
| Cardiac disease, n (ratio) | 1 (0.10) | 12 (0.25) | 0.056 |
| Chronic pulmonary disease | 0 (0.0) | 9 (0.18) | **0.038** |
| Diabetes | 7 (0.35) | 20 (0.41) | 0.609 |
| Chronic kidney disease | 20 (0.100) | 4 (0.08) | **<0.0001** |
| Hypertension | 16 (0.80) | 27 (0.56) | 0.064 |
| Obesity | 1 (0.05) | 7 (0.14) | 0.264 |
| Stroke | 1 (0.05) | 3 (0.06) | 0.842 |
| Charlson Comorbidity index, Mean (SD) | 3.7 (1.5) | 3.1 (2.3) | 0.171 |
| Hospital days | 20.0 (16.2) | 14.7 (17.6) | 0.050 |
| Mortality | 8 (0.4) | 4 (0.08) | **0.002** |
| Severe, n (ratio) | 10 (0.50) | 14 (0.29) | 0.101 |
| **Laboratory Admission** | | | |
| Lymphocytes, cells/µl, Mean (SD) | 715 (449) | 1,026.79 (439) | **0.010** |
| Neutrophils, cells/µl | 5,311 (2,399) | 5,929.63 (3,124) | 0.667 |
| Monocytes, cells/µl | 405 (318) | 381.38 (227) | 0.757 |
| Neutrophil-Lymphocyte Ratio | 11.7 (12.6) | 7.2 (5.4) | 0.095 |
| Platelets, cells/µl | 198,250 (64,921) | 199,729 (73,968) | 0.989 |
| Hemoglobin, g/dL | 12.2 (1.3) | 13.1 (1.8) | **0.022** |
| Hematocrit, (%) | 37.5 (3.7) | 39.1 (5.5) | 0.099 |
| Red Cell Distribution Width, (%) | 13.7 (1.3) | 13.2 (0.9) | 0.154 |
| Creatinine, mg/dL | 2.1 (1.2) | 1.1 (0.9) | **<0.0001** |
| C-Reactive Protein, mg/L | 109.5 (56.5) | 120.7 (94.3) | 0.821 |
| Lactate, mg/dL | 15.2 (10.0) | 13.9 (6.2) | 0.898 |
| D-dimer, µg/mL FEU | 1.7 (2.1) | 2.1 (2.2) | 0.438 |
| Troponin, ng/L | 24.5 (12.8) | 45.4 (116.4) | **0.025** |
| **Laboratory D3** | | | |
| Lymphocytes, cells/µl, Mean (SD) | 503 (257) | 1,509 (816) | **<0.0001** |
| Neutrophils, cells/µl | 6,538 (4,104) | 5,581 (3,015) | 0.533 |
| Monocytes, cells/µl | 353 (201) | 560 (592) | **0.045** |
| Neutrophil-Lymphocyte Ratio | 18.1 (23.9) | 5.21 (4.0) | **<0.0001** |
| Platelets, cells/µl | 231,352 (81,308) | 262,390 (92,191) | 0.172 |

(*Continued*)

**Table 2.** (Continued)

| | SOT (N = 20) | No SOT (N = 48) | P-Value ** |
|---|---|---|---|
| Hemoglobin, g/dL | 11.7 (1.4) | 12.5 (1.7) | 0.082 |
| Hematocrit, (%) | 35.7 (4.0) | 37.7 (5.3) | 0.190 |
| Red Cell Distribution Width, (%) | 13.7 (1.3) | 13.2 (1.0) | 0.188 |
| Creatinine, mg/dL | 2.0 (1.2) | 1.2 (1.6) | **<0.0001** |
| C-Reactive Protein, mg/L | 94.9 (99.4) | 88.8 (82.3) | 0.898 |
| Lactate, mg/dL | 14.2 (4.4) | 18.6 (6.0) | 0.177 |
| D-dimer, μg/mL FEU | 1.5 (1.1) | 2.6 (3.6) | 0.484 |
| Troponin, ng/L | 20.6 (10.0) | 41.7 (80.2) | 0.569 |
| **Laboratory D7** | | | |
| Lymphocytes, cells/μl, Mean (SD) | 685 (317) | 1,366 (728) | **0.002** |
| Neutrophils, cells/μl | 7,365 (3,828) | 7,619 (4,220) | 1.00 |
| Monocytes, cells/μl | 464 (223) | 665 (355) | 0.064 |
| Neutrophil-Lymphocyte Ratio | 14.9 (13.6) | 9.8 (10.8) | 0.072 |
| Platelets, cells/μl | 270,200 (68,033) | 353,000 (141,092) | **0.032** |
| Hemoglobin, g/dL | 11.3 (1.7) | 11.5 (1.6) | 0.736 |
| Hematocrit (%) | 34.9 (4.8) | 35.0 (5.3) | 0.950 |
| Red Cell Distribution Width, (%) | 13.6 (1.5) | 13.1 (1.1) | 0.254 |
| Creatinine, mg/dL | 2.2 (1.2) | 1.2 (0.8) | **0.003** |
| C-Reactive Protein, mg/L | 87.1 (114.7) | 81.2 (72.6) | 0.681 |
| **Hospital Discharge** | | | |
| Lymphocytes, cells/μl, Mean (SD) | 1,109 (644) | 1,958 (723) | **<0.0001** |
| Neutrophils, cells/μl | 8,964 (5,969) | 6,285 (4,566) | 0.084 |
| Monocytes, cells/μl | 775 (512) | 665.81 (449) | 0.311 |
| Neutrophil-Lymphocyte Ratio | 13.1 (19.9) | 3.7 (3.2) | **<0.0001** |
| Platelets, cells/μl | 245,894 (141,791) | 297,380 (130,370) | 0.342 |
| Hemoglobin, g/dL | 11.3 (1.7) | 11.5 (1.6) | **0.002** |
| Hematocrit (%) | 34.9 (5.3) | 35.0 (4.8) | **0.006** |
| Red Cell Distribution Width, (%) | 14.2 (1.7) | 13.2 (1.2) | **0.015** |
| Creatinine, mg/dL | 2.0 (1.0) | 1.0 (0.6) | **<0.0001** |
| C-Reactive Protein, mg/L | 42.0 (44.6) | 33.2 (27.9) | 0.526 |

* Mann-Whitney, t-test or chi-square were applied to determine the P value when comparing groups.

SD (Standard Deviation).

SpO2 (Oxygen Saturation).

SOFA score (Sequential Organ Failure Assessment Score).

recovery or deteriorating clinical course in patients who presented with similar clinical symptoms and disease severity at admission. Patients in the cohort followed either an uncomplicated course, with clinical recovery and short length of stay in the hospital or a complicated course toward critical illness, with ARDS, shock, and renal failure, leading to death or late recovery. Here we report changes in routine hematologic laboratory results that could indicate an unfavorable outcome.

At admission, except for lymphopenia, routine hematologic laboratory tests of the patients were in the range of reference values for neutrophils, monocytes, and platelets. This was also true for hemoglobin, hematocrit and RDW values. When stratifying patients accordingly to their clinical course, those with a deteriorating clinical course presented with lymphopenia, neutrophilia, and a consequent higher NLR. This pattern was clearer in follow-up samples, at

# Relation of Lymphocytes to Transplanted Patients

## Lymphocyte Count on Day 7

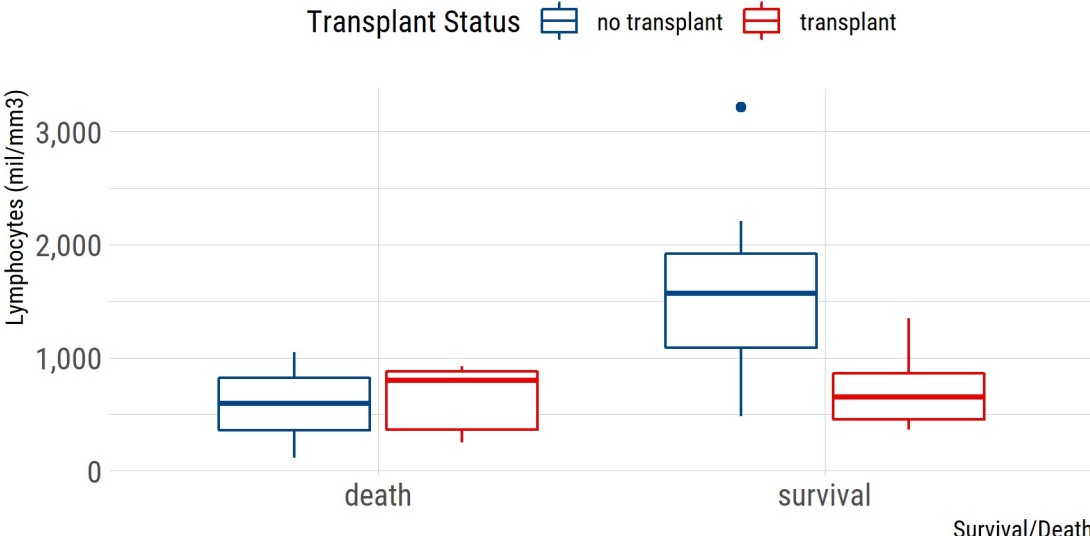

**Fig 2. Lymphocyte's cell counts at day 7 in KTR and non-KTR COVID-19 patients according to disease outcomes.**

D3 and D7, thus paralleling the clinical deterioration, as illustrated by the median time of 3.5 days from hospital admission to ICU, 6 days from hospital admission to mechanical ventilation and of 8 days to the use of vasopressors.

A PCA model using the difference between laboratory results on Day 1 and Day 3 illustrated the potential of routine hematologic changes along with creatinine levels to predict outcomes in moderately ill COVID-19 patients. To develop any score based on this finding will require a much larger number of patients.

Our data confirms and expands previous results in COVID-19 patients and emphasizes the relevance of routine hematologic tests as a simple bed-side parameter for helping the physician to assist COVID-19 patients. In fact, since the first reports on COVID-19 patients' cohorts, lymphopenia has been described as a common finding and predictive of disease outcomes. In retrospective, single time points studies (admission samples), lymphopenia has been associated with decreased SpO2 and disease severity [20,21]. In a retrospective multicenter cohort study evaluating patients discharged alive and those who died, baseline lymphocyte count was significantly higher in survivors than in non-survivors [13]. In a nationwide cohort study performed in Korea, Jongmin Lee et al proposed lymphopenia as a biological predictor of outcomes in COVID-19 patients. They stratified the patients in three propensities cohorts based on lymphocytes cell counts in admission samples, that were related to disease severity and outcomes [22].

Neutrophilia and NLRs have also been related to disease severity and outcomes in COVID-19 patients [23]. In a systematic review and metanalysis Li and coworkers evaluated the predictive values of NLR on disease severity and mortality in COVID-19 patients. Nineteen studies were selected, all but one was retrospective. The AUC of NLR for predicting disease severity was 0.85 (95% CI 0.81–0.88) and for predicting mortality was 0.90 (95% CI 0.87–0.92) [23]. In a prospective study evaluating predictive factors of critical illness in patients admitted with

mild, moderate, severe and critical disease, the NLR had an area under the receiver operating characteristic of 0.849 in the derivation cohort (61 patients) and 0.867 in the validation cohort (54 patients) [11].

Our cohort allows a direct comparison of disease progression and routine laboratory changes in KT recipients and non-KT recipients COVID patients once they were admitted with similar disease severity and were treated following the same clinical protocols.

KT recipients had much higher mortality than non-KT patients in this cohort. Our finding of 40% fatality is in accordance with previous studies that reported mortality ranging from 19% to 50%, although it is important to remark that most of these other studies also included outpatient cases, while our analysis was restricted to hospitalized patients [24–26]. Similar to what we observed in our analysis, in a recent multicenter national study cohort conducted in our country, Requião-Moura et al. evaluated 1,680 KT recipients diagnosed with COVID-19, of whom 65.1% required hospitalization, and the fatality rate of in-hospital patients was 31.6% [27]. The reasons why kidney transplant recipients have presented unfavorable outcomes compared to non-transplanted patients are under investigation. However, the cumulative number of comorbidities, lower baseline kidney function, and the chronic and unavoidable use of immunosuppressive drugs have been pointed out as the main predictors of mortality in this group of patients [28].

One important sign of severity in COVID-19 is acute kidney injury and renal replacement therapy requirement, that was related to a 2.2 chance of death within 60 days in ICU COVID-19 patients [29,30]. Creatinine levels were in fact higher in patients with complicated clinical evolutions than in those with uncomplicated evolutions in our entire cohort. In KT patients, the loss of graft function is one of the main outcomes in COVID-19 patients who have been studied [25,26]. Indeed, in our cohort, renal replacement therapy (RRT) was needed in 40% of KT patients, in agreement with a large Brazilian study of COVID-19 in KT patients reporting that 45% of patients needed RRT [6]. Moreover, acute kidney injury was reported in 45% of hospitalized patients in a French cohort and in 52% of 144 patients of a multicenter cohort of KT recipients with COVID-19 [26,31].

Routine hematologic changes in our cohort shows lower lymphocyte levels and a trend toward higher NLR ratios in the KT recipients compared with non-KT recipients. Unexpectedly, lymphopenia was not predictive of complicated outcomes among KT recipients. The lymphocyte cell counts were quite similar in KT recipients with complicated and uncomplicated COVID-19 clinical courses. This might be interpreted as a direct effect of immunosuppressive therapy in these patients, which would blunt the effect of SARS-CoV2 infection. This result is in contrast with TANGO Consortium, which found statistically significant differences in lymphocyte count between survivors and non survivors [26]. Ongoing evaluation of lymphocytes populations and functions will be important to unravel differences among patients with distinct clinical courses. Neutrophils and NLR in KT recipients were higher in those patients deteriorating to critical illness only at hospital discharge. Thus, among KT COVID-19 patients, white blood cells fractions were not good parameters to distinguish patients with complicated and uncomplicated clinical course except at hospital discharge. The effects of other variables, such as concurrent infections and therapeutic interventions, should be considered as possible confounders for this finding. Hematocrit and hemoglobin were lower at hospital discharge for those KT recipients with critical illness, in this case, like the general population in the cohort.

## This work has strengths and limitations

As a strength, it is a prospective cohort of patients with laboratory samples obtained at various time points. Most studies evaluating routine hematologic tests are retrospective and many are

based on admission samples. The narrow scope of clinical severity, representing a typical patient admitted in general COVID-19 wards, is relevant for the search of parameters related to clinical deterioration or recovery. The presence of KT recipients among the cohort allowed a direct comparison of clinical course with non-KT recipients. There are several limitations to the study. This is a single center cohort, and the conclusions may not be generalizable to other hospitals. The small sample size did not allow us to perform conclusive analyses in the subgroups of patients. KT recipients are restricted to one transplantation center and might not represent the general KT population.

In conclusion, routine hematologic tests seem to be a useful parameter to assist the physician in the clinical care of patients admitted in the hospital wards with COVID-19 disease. Lymphocyte cell counts and NLRs are easily accessed laboratory tests to monitor deteriorating or recovering clinical courses. KT recipient COVID-19 patients had a dramatically higher lethality ratio than non-KT patients. Surprisingly, lymphocyte cell counts are not associated with outcomes in this group. Ongoing studies of inflammatory mediators, lymphocyte subpopulations and functions and the proteomics of peripheral blood mononuclear cells in KT recipient COVID-19 patients may contribute to increased understanding of COVID-19 pathogenesis in the clinical setting.

## Supporting information

**S1 Fig. Demographic data of the COVID-19 cohort.**
(DOCX)

**S2 Fig. A principal components model showing the separation in values for each of the laboratory tests between day 1 and day 3.**
(DOCX)

**S3 Fig.** CRP (A) and Creatinine (B) levels in COVID-19 patients accordingly to the clinical course and outcomes.
(DOCX)

**S1 Table. Routine laboratory tests of the cohort according to the clinical course and outcomes of the disease.**
(DOCX)

**S2 Table. Demographic, clinical and laboratory results of non-KT recipients COVID-19 patients accordingly to the clinical course of the disease.**
(DOCX)

**S3 Table. Demographic, clinical and laboratory results of KT recipients COVID-19 patients accordingly to the clinical course of the disease.**
(DOCX)

**S4 Table. Raw data required to replicate the results of your study.**
(XLSX)

## Acknowledgments

We thank Professor José Medina-Pestana and Professor Lúcio R. Requião-Moura from Nephrology Division at Federal University of São Paulo and Hospital do Rim for helpful discussion related to kidney transplantation.

## Author Contributions

**Conceptualization:** Paula M. Peçanha-Pietrobom, Paulo R. Abrão Ferreira, Marcelo N. Burattini, Nancy Bellei, Jaquelina Sonoe Ota-Arakaki, Reinaldo Salomao.

**Data curation:** Paula M. Peçanha-Pietrobom, Giuseppe Gianini Figueirêdo Leite, Paulo R. Abrão Ferreira, Nancy Bellei, Jaquelina Sonoe Ota-Arakaki, Reinaldo Salomao.

**Formal analysis:** Paula M. Peçanha-Pietrobom, Giuseppe Gianini Figueirêdo Leite, James Hunter, Marcelo N. Burattini, Nancy Bellei, Jaquelina Sonoe Ota-Arakaki, Reinaldo Salomao.

**Funding acquisition:** Reinaldo Salomao.

**Investigation:** Paula M. Peçanha-Pietrobom, Paulo R. Abrão Ferreira, Nancy Bellei, Jaquelina Sonoe Ota-Arakaki, Reinaldo Salomao.

**Methodology:** Paula M. Peçanha-Pietrobom, James Hunter, Paulo R. Abrão Ferreira, Marcelo N. Burattini, Jaquelina Sonoe Ota-Arakaki, Reinaldo Salomao.

**Project administration:** Paulo R. Abrão Ferreira, Jaquelina Sonoe Ota-Arakaki, Reinaldo Salomao.

**Resources:** Paula M. Peçanha-Pietrobom, Paulo R. Abrão Ferreira, Jaquelina Sonoe Ota-Arakaki, Reinaldo Salomao.

**Software:** James Hunter.

**Supervision:** Jaquelina Sonoe Ota-Arakaki, Reinaldo Salomao.

**Validation:** James Hunter, Reinaldo Salomao.

**Visualization:** James Hunter, Reinaldo Salomao.

**Writing – original draft:** Paula M. Peçanha-Pietrobom, Giuseppe Gianini Figueirêdo Leite, James Hunter, Paulo R. Abrão Ferreira, Jaquelina Sonoe Ota-Arakaki, Reinaldo Salomao.

**Writing – review & editing:** Paula M. Peçanha-Pietrobom, James Hunter, Paulo R. Abrão Ferreira, Jaquelina Sonoe Ota-Arakaki, Reinaldo Salomao.

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
