## [Decision Letter · Decision Letter 0]

2 Aug 2021

PONE-D-21-15896

Clinical course of COVID-19 patients – with and without kidney

transplantation – admitted at the hospital with moderate disease is

mirrored by hematologic routine laboratory tests and deeply influenced by

transplant history

PLOS ONE

Dear Dr. Salomao,

Thank you for submitting your manuscript to PLOS ONE. After careful consideration, we feel that it has merit but does not fully meet PLOS ONE’s publication criteria as it currently stands. Therefore, we invite you to submit a revised version of the manuscript that addresses the points raised during the review process.

We look forward to receiving your revised manuscript.

Kind regards,

Chiara Lazzeri

Academic Editor

PLOS ONE

Additional Editor Comments (if provided):

Reviewers' comments:

Reviewer's Responses to Questions

**Comments to the Author**

1. Is the manuscript technically sound, and do the data support the conclusions?

Reviewer #1: Partly

Reviewer #2: Yes

2. Has the statistical analysis been performed appropriately and rigorously? 

Reviewer #1: N/A

Reviewer #2: Yes

3. Have the authors made all data underlying the findings in their manuscript fully available?

Reviewer #1: Yes

Reviewer #2: Yes

4. Is the manuscript presented in an intelligible fashion and written in standard English?

Reviewer #1: Yes

Reviewer #2: No

5. Review Comments to the Author

Reviewer #1: The authors made a great effort to support their thesis. They have great credit for following their population prospectively. However, the blood chemistry parameters they consider are also the first to change over the course of any infection and inflammation. There is no doubt that these changes are part of the course of COVID disease, but their ability to predict outcome is rather weak, correlation with the specific pathology under consideration should be improved.

I suggest to the authors to improve the study design, not limiting themselves only to describing but trying to correlate what was observed to the specific pathology in a stronger and more unambiguous way. I also suggest that we organize the discussion better to make it more linear and less confusing. Better to stick to the description of the results and their interpretation more concisely.

Reviewer #2: Could benefit from minor sytlistic/language revision for clarity sake. The findings of persistent elevation in NLR correlating with mortality is not unsurprising and did not change management, but the data may still be valuable for risk profiling.

6. PLOS authors have the option to publish the peer review history of their article (what does this mean?). If published, this will include your full peer review and any attached files.

Reviewer #1: No

Reviewer #2: No

---

## [Author Response · Author response to Decision Letter 0]

18 Sep 2021

We thank the referees’ comments, which are responded point by point below.

1. Is the manuscript technically sound, and do the data support the conclusions?

Reviewer #1: Partly

Response: we have reviewed the manuscript and hope to have improved its quality and presentation.

Reviewer #2: Yes

2. Has the statistical analysis been performed appropriately and rigorously?

Reviewer #1: N/A

Reviewer #2: Yes

3. Have the authors made all data underlying the findings in their manuscript fully available?

Reviewer #1: Yes

Reviewer #2: Yes

4. Is the manuscript presented in an intelligible fashion and written in standard English?

Reviewer #1: Yes

Reviewer #2: No

Response: we have reviewed the manuscript and hope to have improved its quality and presentation.

 5. Review Comments to the Author

Reviewer #1: The authors made a great effort to support their thesis. They have great credit for following their population prospectively. However, the blood chemistry parameters they consider are also the first to change over the course of any infection and inflammation. There is no doubt that these changes are part of the course of COVID disease, but their ability to predict outcome is rather weak, correlation with the specific pathology under consideration should be improved.

I suggest to the authors to improve the study design, not limiting themselves only to describing but trying to correlate what was observed to the specific pathology in a stronger and more unambiguous way. I also suggest that we organize the discussion better to make it more linear and less confusing. Better to stick to the description of the results and their interpretation more concisely.

Response: Thank you for your comments.

We have completely revised the manuscript, focusing on the cohort description and the main results obtained. To make the message clearer, we modified the tables that were in the main text and those presented as supplementary.

For some comparisons (as in supplementary table 1) we changed the statistical approach to compare between patients deteriorating to critical illness and those with clinical recovery, and among the first ones between those surviving and not surviving. Furthermore, we added a Principal Components Analysis (PCA) in order to identify the set of variables that most correlates with outcomes.

Results are presented focusing on the routine hematologic changes and clinical outcomes; the issue of kidney transplant recipients (KTR) COVID-19 patients is presented more concisely.

The discussion was reorganized, as suggested. It is more concise (original version – 2167 words; revised version – 1489 words) and focused on the results.

The quality of figures is also improved.

The file “text marked with tracks” highlight part of these changes. Part of the texts that were removed could not be marked.

We hope to have improved the quality of the manuscript and attended your expectations.

Reviewer #2: Could benefit from minor sytlistic/language revision for clarity sake. The findings of persistent elevation in NLR correlating with mortality is not unsurprising and did not change management, but the data may still be valuable for risk profiling.

Response: We thank you for the comments. Please, see the above response to the reviewer 1. 

The text was reviewed, it is more focused, concise and, hopefully, also clear. 

We hope to have improved the quality of the manuscript and attended your expectations.

---

## [Editor Report · Decision Letter 1]

11 Oct 2021

The clinical course of hospitalized moderately ill COVID-19 patients is mirrored by routine hematologic tests and influenced by renal transplantation.

PONE-D-21-15896R1

Dear Dr. Salomao,

We’re pleased to inform you that your manuscript has been judged scientifically suitable for publication and will be formally accepted for publication once it meets all outstanding technical requirements.

Kind regards,

Chiara Lazzeri

Academic Editor

PLOS ONE
---

## [Editor Report · Acceptance letter]

4 Nov 2021

PONE-D-21-15896R1 

The clinical course of hospitalized moderately ill COVID-19 patients is mirrored by routine hematologic tests and influenced by renal transplantation. 

Dear Dr. Salomao:

I'm pleased to inform you that your manuscript has been deemed suitable for publication in PLOS ONE. Congratulations! Your manuscript is now with our production department. 

Kind regards, 

on behalf of

Dr. Chiara Lazzeri 

Academic Editor

PLOS ONE